# Genomic dissection of methane emission traits in cattle: A meta-GWAS and heritability analysis across populations

Sare Golpasand, Navid Ghavi Hossein-Zadeh ◉*, Shahrokh Ghovvati ◉

Department of Animal Science, Faculty of Agricultural Sciences, University of Guilan, Rasht, Iran

* nhosseinzadeh@guilan.ac.ir, navid.hosseinzadeh@gmail.com

## Abstract

Enteric methane emissions from ruminants represent a significant contributor to agricultural greenhouse gases, necessitating precise genetic tools to guide mitigation strategies. This study aimed to identify genomic regions and estimate heritability parameters associated with methane-related traits in cattle through an integrated meta-analytical framework. The meta-analysis of the genome-wide association studies (meta-GWAS) was carried out with the METAL software, combining SNP level data extracted from published studies. Simultaneously, a distinct random effects meta-analysis of genomic and pedigree-based heritability estimates was performed using Comprehensive Meta-Analysis software. Functional analysis of the post-GWAS, including: Gene Ontology, KEGG, and network-based enrichment analysis, was also performed to describe the biological context of significant genes. The meta-GWAS identified 74 significant SNPs that were significant for the traits of methane, which are related to 113 candidate genes. Functional enrichment analyses revealed pathways related to metabolism, immune response, ion transport, and host–microbiome interactions. The KEGG metabolic pathway emerged as a highly enriched term, encompassing key genes such as: *ALDH7A1*, *CYP51A1*, *P4HA2*, and *SHPK*, which are involved in amino acid catabolism, lipid processing, and redox regulation functions critical to energy balance and digestive efficiency. Network analysis with Cytoscape has revealed *TRPV3, TRPV1, ANK3, PKD2* and *SHPK* as network hub genes. Heritability meta-analysis indicated that methane production exhibited the moderate genomic ($h^2 = 0.296$) and pedigree-based ($h^2 = 0.299$) heritability estimations, and methane yield was also found to have moderate and high heritability. The findings highlight the potential for methane-related traits as viable targets for genetic selection. This research demonstrates the value of integrating functional genomics and quantitative genetic approaches to enhance understanding of the biological and heritable components of methane emissions, providing a robust foundation for an environmentally sustainable livestock breeding program.

**Data availability statement:** All relevant data are within the manuscript and its Supporting information files.

**Funding:** The author(s) received no specific funding for this work.

**Competing interests:** The authors have declared that no competing interests exist.

## Introduction

With the growing global concern about climate change, methane has emerged as a critical target in mitigation strategies. It has a high global warming potential and a relatively short atmospheric lifespan that makes it a unique point of focus to achieve accelerated benefits of climate. Despite technological progress in controlling industrial methane emissions, enteric fermentation in ruminant livestock remains a major contributor to global methane emissions [1]. Over the past 130 years, methane emissions from livestock, particularly from ruminants, particularly cattle and buffalo have increased more than fourfold, and the largest contributors were non-dairy and dairy cattle [1]. Thus, effective mitigation strategies in the livestock industry must integrate nutritional interventions, genetic selection, and improvements in production efficiency to reduce environmental impact without affecting animal health, productivity, or system sustainability [2–7].

Genome-wide association studies (GWAS) have become a powerful approach to genetic variants and markers associated with economically important and complex traits in livestock, particularly in cattle [8–10]. Statistical modeling and high-density single-nucleotide polymorphism (SNP) arrays have enabled the identification of genotype-phenotype associations across populations, enhancing our knowledge about the molecular architecture of complex traits Moreover, GWAS summary statistics are useful as inputs to meta-analyses, fine-mapping and the generation of genomic prediction models [11–13].

In recent years, GWAS have been increasingly applied to sustainability-related traits such as: disease resistance, heat tolerance, feed efficiency, and methane emission in ruminants [14–16]. GWAS also provide a genetic basis to refine the selection indices and improve the precision of genomic evaluations in livestock breeding programs by discovering significant SNPs [17]. Several GWAS studies have been conducted to investigate methane-related traits in livestock [16,18–20].

In addition to the identification of significant genomic regions, a precise and quantitative understanding the genetic parameters underlying methane emission-related traits, including their heritability and genetic correlations with key economic traits, is essential for developing sustainable breeding strategies. Recent studies in ruminants confirm that methane emissions show significant heritable variation without having an adverse impact on productivity [21]. Methane emission is a measurable environmental trait that can be incorporated directly or indirectly into selection indices. Thus gradual minimization of the environmental footprint of livestock production is a viable and sustainable approach to environmental mitigation through genetic selection [22]. However, heritability estimates for methane traits have been reported to differ widely because of the variation in population size, breed composition, measurement methods, and models of analysis. Integrating results through meta-analysis provides a robust statistical framework for synthesizing these heterogeneous findings. Specifically, random-effects models consider both within- and between-study variation, and the estimates are more representative of actual genetic effects in different populations [23–26]

Additionally, integrative genomic methods that involve the combination of GBLUP and GWAS have been found to be useful in estimating heritability and also identifying candidate genes. For instance, genomic analyses in Nellore cattle have identified several genes that regulate rumen microbiota, lipid metabolism, and fatty acid biosynthesis that contribute to genetic variation in methane emissions [18].

Integrating GWAS results with reliable heritability estimates, especially in the context of genomic selection, has a high potential for developing selection indices and breeding strategies that would reduce greenhouse gas emissions in livestock [27,28]. This effort is reinforced by meta-analysis, which combines results from independent studies, thereby increasing the statistical power to detect actual genetic associations and improving the accuracy and reliability of results [13,29]. The recent methodological developments in statistical for genomic meta-analysis have significantly increased the ability to detect subtle genetic effects and have provided to clarify the missing heritability that is frequently lost in individual studies [30]. Furthermore, large-scale international collaborations and research consortia, which increase sample sizes and include genetically diverse populations have enhanced the discovery of novel loci and improved our understanding of complex traits [31]. Recognizing the urgent necessity of environmentally sustainable livestock systems, the current study combines two complementary methods, a meta-analysis of published GWAS findings and a random-effects meta-analysis of genomic and pedigree-based heritability estimates. This integrative framework enhances the genetic knowledge of the methane-related traits in cattle by synthesizing information in a wide range of populations and study designs, which increases statistical power, decreases study-specific bias, and improves the ability to identify important genomic regions associated to methane emissions. The findings of this study are useful for designing environmentally sustainable breeding strategies to reduce the environmental impact of livestock production.

## Materials and methods

### Search strategy and data selection criteria

A systematic and comprehensive review of the literature on GWAS, genomic heritability, and pedigree-based heritability studies related to methane-related traits in cattle was performed based on the PRISMA (Preferred Reporting Items for Systematic Reviews and Meta-Analyses) guidelines [32] (Figs 1 and 2 and S1 Checklist).

The last searches of the database were carried out on 30 September 2025 to identify all relevant studies. Literature searches were performed using multiple databases, including Google Scholar (https://scholar.google.com), PubMed (https://pubmed.ncbi.nlm.nih.gov), Scopus (https://www.scopus.com), Web of Science (https://www.webofscience.com), and ResearchGate (https://www.researchgate.net) databases.

To guide the search strategy and ensure a structured and focused retrieval process, the Population, Intervention, Comparison, and Outcome (PICO) framework was adopted [33]. Three distinct categories of analysis were considered under this framework; For GWAS meta-analysis, the population included dairy and beef cattle breeds evaluated for methane-related traits. The intervention involved GWAS employing SNP marker arrays. Comparisons were made across different SNP markers, cattle populations, or multiple studies. The outcome of interest was the identification of SNPs significantly associated with methane-related traits. The second category focused on studies reporting genomic heritability estimates. The population consisted of cattle breeds assessed for methane-related traits. Interventions involved genomic heritability estimation using SNP-based genomic relationship models, such as genomic best linear unbiased prediction (GBLUP), BayesC, or single-step genomic approaches. Comparisons were drawn across different breeds, estimation models, or population structures. The outcome was the estimation of genomic heritability ($h^2_g$) for methane production, methane yield, or associated traits. In the pedigree-based heritability analysis, the target population included cattle breeds with available phenotypic data for methane-associated traits. Studies estimating pedigree-based heritability using Bayesian inference and restricted maximum likelihood (REML) estimation methods in animal models with additive genetic relationship matrices have been included in the interventions. Comparisons were made across different breeds, statistical models, or estimation methodologies. The outcome was pedigree-based heritability estimates ($h^2_a$) for methane production and related traits.

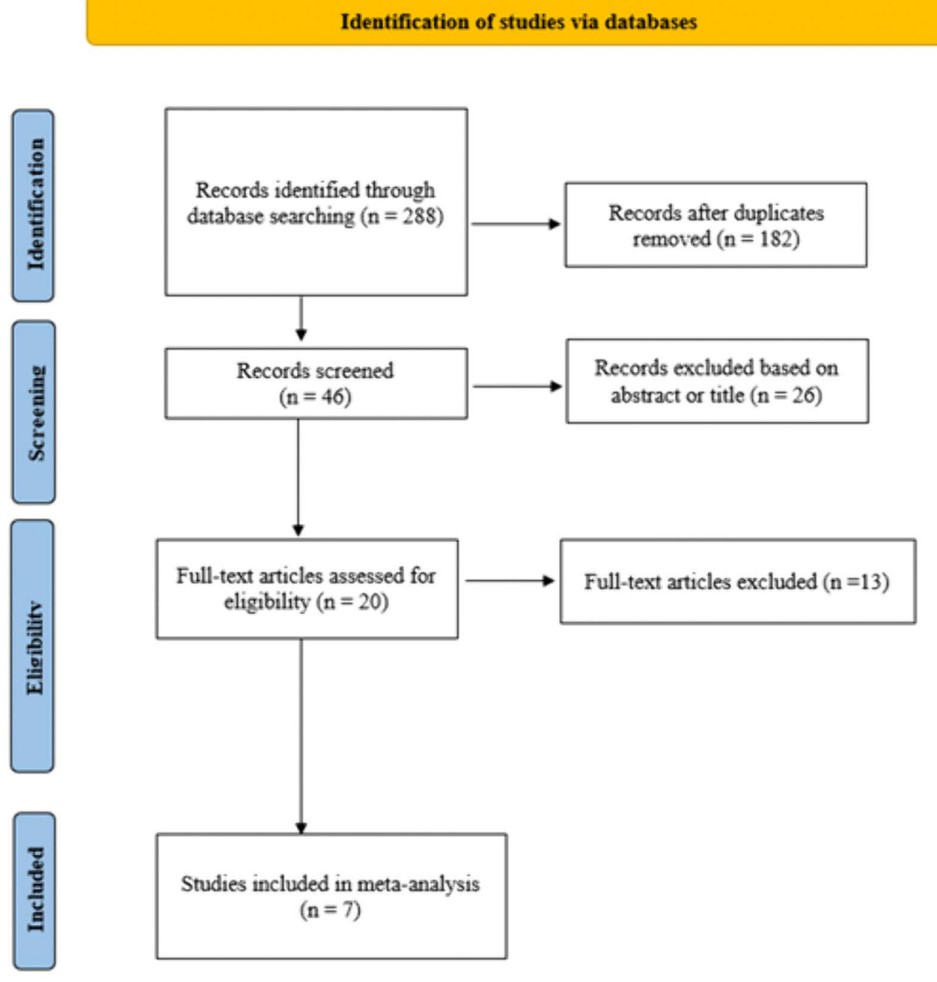

Source: Page MJ, et al. BMJ 2021;372:n71. doi: 10.1136/bmj.n71.

**Fig 1. Flow diagram representing the studies incorporated into the GWAS meta-analysis by the PRISMA approach.**

Additionally, the following strategically combined search terms ensured comprehensive retrieval of relevant studies:

• ("GWAS" OR "genome-wide association") AND ("methane production" OR "methane emission" OR "methane yield" OR "residual methane" OR "methane intensity" OR "methane concentration" OR "enteric methane") AND ("cattle" OR "cow" OR "bovine")

• ("genomic heritability" OR "heritability" OR "genetic parameter") AND ("methane production" OR "methane emission" OR "methane yield" OR "residual methane" OR "methane intensity" OR "methane concentration" OR "enteric methane") AND ("cattle" OR "cow" OR "bovine")

No restrictions were imposed on the publication date to ensure maximal coverage of all relevant studies.
In total, the titles of 46 articles were screened, of which 20 articles were assessed in full text for eligibility
based on the use of GWAS for methane-related traits and the availability of SNP-level results, including genomic

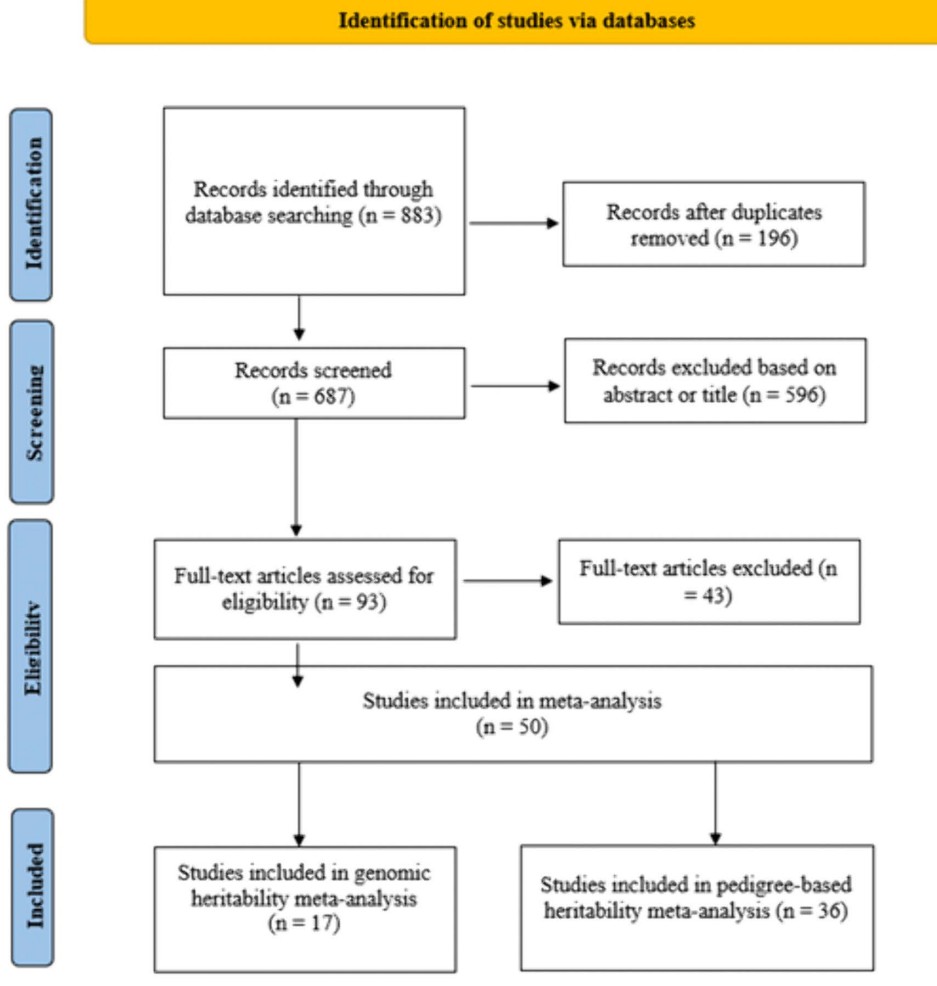

**Fig 2. Flow diagram representing the studies incorporated into the heritability meta-analysis by the PRISMA approach.**

coordinates and P-values. After a thorough evaluation, 7 articles met the inclusion criteria and were retained for the meta-analysis. From the identified studies, data including study title, SNP identifier (marker name), chromosome number, genomic position, P-value, and sample size were extracted and compiled into an Excel file. Additionally, 687 articles related to the heritability estimates for methane traits were initially screened. Based on the title review, 93 articles were deemed relevant and were assessed in full text. After excluding 43 studies due to insufficient or incomplete information, 50 articles were ultimately included for heritability estimates in the meta-analysis. Additional file 1 S1 Table provides a summary of the total number of records from studies included in the heritability meta-analysis. It presents key metadata for each study, including the reference, country or region of origin, cattle breed, type of genetic relationship matrix employed (pedigree or genomic-based), statistical method used, and the analytical model implemented for heritability estimation.

## GWAS meta-analysis

For the meta-analysis, marker names, P-values, and sample sizes extracted from the selected GWAS articles were compiled into text files compatible with METAL software (version 2011-03-25), a command-line tool for GWAS meta-analytic studies [34]. The analysis was conducted using the sample-size–weighted Z-score method implemented in METAL. Due to the lack of consistent reference allele information across the included studies, it was assumed that the effect directions were consistent for all markers. Following the meta-analysis, genome-wide significance was assessed using the Bonferroni correction. The significance threshold was determined by dividing the conventional alpha level ($\alpha = 0.05$) by the total number of SNPs included in the meta-analysis.

## Downstream functional annotation, gene ontology, and network analysis

Significant SNPs identified through meta-analysis were examined in order to determine any genes that could be related to methane emission traits. Genomic position and variant identifiers and data were retrieved and confirmed by Ensembl BioMart (https://asia.ensembl.org/info/website/index. html) and Variant Effect Predictor (VEP) database (https://asia.ensembl.org/Tools/VEP). Additionally, genes associated with SNPs reported in previous GWAS studies were incorporated to strengthen the candidate gene list. This integrative approach resulted in the identification of 113 genes (S2 Table), forming the foundation for subsequent post-GWAS analyses. Network analysis employs mathematical and computational methodologies to investigate the structure, function, and dynamics of interconnected systems. This approach, commonly referred to as network science, focuses on the systematic analysis of complex systems composed of interacting elements. These elements, referred to as nodes, are connected through edges or links that represent various types of relationships, collectively forming a network [35]. Protein-protein interaction (PPI) network analysis enables the exploration of functional relationships among proteins and provides insight into their roles within broader biological systems. To explore the interactions among the genes, a PPI network was constructed using the STRING database (https://string-db.org/) and visualized with Cytoscape software [36]. The CytoHubba plug-in was applied using the maximal clique centrality (MCC) algorithm to identify the five key hub genes based on their degree of centrality within the network. Functional enrichment analysis was conducted using the ClueGO and CluePedia plugins within Cytoscape [37,38]. The analysis included all genes associated with significant SNPs and was carried out across the three Gene Ontology (GO) categories: biological process (BP), molecular function (MF), and cellular component (CC). In order to confirm and complement these results, a secondary GO-based enrichment analysis was conducted using the BiNGO plug-in in Cytoscape [39]. The default parameters of BiNGO were used, using the same gene list and GO categories (BP, MF, CC), with a significance threshold of $P < 0.05$ after FDR correction. The created hierarchical GO tree using BiNGO was color-coded based on the statistical significance to facilitate interpretation of enriched biological functions.

To further refine the network analysis, the genes with significant SNPs were mapped to their human orthologs, and a gene network analysis for 86 genes was performed with GeneMANIA in Cytoscape [40]. This tool constructs a comprehensive gene interaction network using various data sources, including co-expression, co-localization, physical interactions, shared protein domains, predicted functional relationships, and literature-based associations to construct a comprehensive gene interaction network. Finally, to obtain a broader understanding of gene function, enrichment analyses were performed using the DAVID web tool (https://davidbioinformatics.nih.gov/) that enables enrichment analysis across three main functional categories: biological processes (BP), cellular components (CC), and molecular functions (MF). In addition, pathway enrichment based on the KEGG database was conducted to identify key biological and metabolic pathways potentially involved in methane emission.

## Meta-analysis of genomic and pedigree-based heritability estimates

Genomic and pedigree-based heritability estimates were meta-analyzed separately using Comprehensive Meta-Analysis (CMA) software version 3.0 [41]. The analysis of genomic heritability estimates was limited to two traits: methane

production (METP) and methane concentration (METC). Eligible studies were systematically searched and identifiable estimates of heritability and their standard errors were obtained. In cases where standard errors for heritability estimates were not reported in the articles, these values were estimated using the combined variance method [42]. The formula for calculating the predicted standard error for articles that lacked this information is defined as follows:

$$SE_{ij} = \sqrt{\frac{\left(\frac{\sum_{k=1}^{K} s_{ik}^2 n_{ik}^2}{\sum_{k=1}^{K} n_{ik}}\right)}{n'_{ij}}}$$

In this equation, $SE_{ij}$ denotes the estimated standard error associated with the genetic parameter reported for the $i$th trait in the $j$th article, in which the original standard error was not provided. $S_{ik}$ represents the observed standard error for the estimate of the $i$th trait in the $k$th article, which includes both the parameter estimate and its standard error. $n_{ik}$ refers to the number of records used to derive the parameter estimate in the $k$th study, and n'ij is the number of records used to predict the published parameter estimate for the $i$th trait in the $j$th article that has not reported the standard error.

The extracted data were subsequently imported into CMA for statistical analysis. Given the expected heterogeneity across studies, a random-effects model was employed. Following the meta-analysis, the pooled heritability estimate and its associated standard error were calculated.

## Evaluation of publication bias

To evaluate the presence of publication bias, Egger's linear regression test was employed. When statistical evidence of bias was detected (P<0.10), the Trim-and-Fill method was applied to detect the number of potentially missing studies [43]. A Funnel plot was also generated and visually inspected to assess the symmetry of the effect size distribution. The results were analyzed and interpreted in the context of between-study heterogeneity and overall effect size estimation. After estimating the number of missing studies, the imputed values were incorporated into the model to recalculate the weighted mean effect size and its associated variance. It should be noted that in the presence of significant heterogeneity across studies (Q test with P<0.10), the validity of publication bias tests may be compromised, potentially leading to false positive claims [44].

## Results

### GWAS meta-analysis

This meta-analysis integrated genomic data, specifically SNP positions, chromosomal locations, and p-values, extracted from relevant studies identified through a systematic literature review (Table 1).
Due to the lack of reported SNP effect sizes in the original studies, the combined statistical analyses were based exclusively on the p-values derived from each study. After the meta-analysis, 74 SNPs were found to be statistically significant (P-value<0.0003), and it was found that they are association with methane emission traits. The number of significant SNPs per chromosome is summarized in Table 2, offering an overview of their genome-wide distribution patterns associated with methane emission traits.

The Table 3 provides detailed information on the identified SNPs, including their chromosomal positions and statistical significance.

Additionally, Z-score values, derived from the transformation of combined P-values, represent both the magnitude and directionality of the aggregated SNP effects across the analyzed studies.

### Functional annotation, gene ontology, and network analysis

The meta-analysis resulted in the identification of 113 genes associated with significant SNPs. To explore the functional roles of these genes, pathway annotations were retrieved from the DAVID database and categorizedinto three major

**Table 1. Characteristics of GWAS papers included in databases for performing a meta-analysis.**

| Reference | Number of Animals | Number of Significant SNPs | Breed |
|---|---|---|---|
| Souza et al. [18] | 743 | 11 | Nellore cattle |
| Lekamp et al. [16] | 802 | 28 | crossbred beef steers |
| Manzanilla-Pech et al. [45] | 1844 | 21 | Danish Holstein cattle |
| Jalil Sarghale et al. [19] | 150 | 9 | Iranian Holstein cattle |
| Jalil Sarghale et al. [46] | 150 | 5 | Iranian Holstein cattle |
| Calderon-Chagoya et al. [47] | 280 | 46 | Dairy cattle (Simmental, Holstein, or Brown Swiss) |
| Pszczola et al. [20] | 287 | 50 | Polish Holstein-Friesian |

SNP: Single Nucleotide Polymorphism.

**Table 2. The number of significant SNPs per chromosome.**

| Chromosome number | Length of chromosome (bps) | Number of SNPs |
|---|---|---|
| 1 | 158,534,110 | 8 |
| 2 | 136,231,102 | 1 |
| 3 | 121,005,158 | 2 |
| 4 | 120,000,601 | 2 |
| 5 | 120,089,316 | 5 |
| 6 | 117,806,340 | 1 |
| 7 | 110,682,743 | 7 |
| 8 | 113,319,770 | 1 |
| 9 | 105,454,467 | 2 |
| 10 | 103,308,737 | 10 |
| 11 | 106,982,474 | 1 |
| 12 | 87,216,183 | 3 |
| 13 | 83,472,345 | 3 |
| 14 | 82,403,003 | 2 |
| 15 | 85,007,780 | 2 |
| 16 | 81,013,979 | 1 |
| 17 | 73,167,244 | 6 |
| 18 | 65,820,629 | 1 |
| 19 | 63,449,741 | 3 |
| 20 | 71,974,595 | 3 |
| 23 | 52,498,615 | 2 |
| 24 | 62,317,253 | 2 |
| 28 | 45,940,150 | 3 |

GO domains: BP, CC, and MF. The results indicated that the candidate genes are involved in a wide range of biological functions and pathways. Detailed functional annotations for each gene, including gene ID, gene name, and associated GO terms, are provided in S3 Table, which provides additional information on the different biological functions of individual genes, especially those are not present in any gene interaction networks or functional clustering analyses. In this study, a topological analysis of significant nodes in an interactome network was conducted by CytoHubba plug-in, which combines topological analysis algorithms. Among the eleven topological analysis methods implemented in CytoHubba, MCC was found to be the most accurate in identifying essential proteins [48] and hence it was used to identify essential nodes.

 

**Table 3. Detailed information of significant SNPs identified in GWAS Meta-analysis.**

| Marker name | CHR | Genomic position | Allele 1 | Allele 2 | Weight* | Z-score | P-value |
|---|---|---|---|---|---|---|---|
| 1_54481566 | 1 | 54481566 | T | G | 150 | 5.137 | 2.79E-07 |
| BovineHD0100039506 | 1 | 138388705 | T | G | 280 | 4.553 | 5.30E-06 |
| BovineHD0100033007 | 1 | 116746240 | A | C | 280 | 4.36 | 1.30E-05 |
| BovineHD0100026381 | 1 | 92730269 | T | C | 280 | 4.107 | 4.00E-05 |
| BovineHD0100035695 | 1 | 126404986 | A | G | 280 | 4.096 | 4.20E-05 |
| BovineHD0100024203 | 1 | 84485319 | A | C | 280 | 3.927 | 8.60E-05 |
| BovineHD0100022218 | 1 | 77090546 | T | C | 280 | 3.643 | 0.00027 |
| BovineHD0100025559 | 1 | 89815855 | A | G | 280 | 3.624 | 0.00029 |
| ARS-BFGL-NGS-116634 | 2 | 107318080 | A | G | 150 | 4.866 | 1.14E-06 |
| rs110220315 | 3 | 86090597 | A | G | 2406 | 6.75 | 1.48E-11 |
| rs110058749 | 3 | 88164394 | G | A | 2406 | 6.513 | 7.38E-11 |
| BovineHD0300026630 | 3 | 92487386 | A | G | 280 | 3.867 | 0.00011 |
| 4:115131249 | 4 | 115131249 | G | C | 150 | 6.351 | 2.14E-10 |
| rs134296722 | 4 | 104579265 | T | C | 1604 | 5.197 | 2.02E-07 |
| BovineHD0400033637 | 4 | 115960201 | A | G | 280 | 3.633 | 0.00028 |
| rs109244569 | 5 | 21516232 | T | G | 2406 | 6.935 | 4.06E-12 |
| rs110309656 | 5 | 21475995 | C | A | 2406 | 6.935 | 4.06E-12 |
| rs42740586 | 5 | 1870477 | G | A | 2406 | 6.585 | 4.56E-11 |
| rs137645685 | 5 | 60495101 | G | A | 2406 | 6.516 | 7.22E-11 |
| BovineHD0500011829 | 5 | 44357278 | A | C | 280 | 4.265 | 2.00E-05 |
| BovineHD0600033597 | 6 | 117942392 | T | C | 150 | 5.304 | 1.13E-07 |
| rs43508672 | 7 | 22191462 | A | G | 2406 | 7.581 | 3.44E-14 |
| rs43508661 | 7 | 22204260 | A | G | 2406 | 7.581 | 3.44E-14 |
| rs43508667 | 7 | 22197623 | A | G | 2406 | 7.581 | 3.44E-14 |
| rs29023390 | 7 | 22202959 | G | A | 2406 | 7.581 | 3.44E-14 |
| rs43508669 | 7 | 22195580 | T | C | 2406 | 7.581 | 3.44E-14 |
| rs43509246 | 7 | 22328341 | C | T | 2406 | 7.581 | 3.44E-14 |
| rs43141114 | 7 | 22268814 | G | A | 2406 | 7.581 | 3.44E-14 |
| rs41621748 | 8 | 56479469 | G | A | 2406 | 6.503 | 7.87E-11 |
| BovineHD080011543 | 8 | 38725165 | A | C | 150 | 4.903 | 9.43E-07 |
| rs43605790 | 9 | 89807465 | G | A | 2406 | 9.723 | 2.41E-22 |
| BovineHD0900021600 | 9 | 77448787 | A | G | 280 | 4.085 | 4.40E-05 |
| rs132818032 | 10 | 67219330 | A | G | 743 | 5.508 | 3.63E-08 |
| rs109315063 | 10 | 67275755 | G | A | 743 | 5.508 | 3.63E-08 |
| rs134213606 | 10 | 67236195 | G | A | 743 | 5.508 | 3.63E-08 |
| rs135715112 | 10 | 67262107 | G | A | 743 | 5.508 | 3.63E-08 |
| rs135939893 | 10 | 67227401 | A | G | 743 | 5.508 | 3.63E-08 |
| rs133164059 | 10 | 67254747 | G | A | 743 | 5.508 | 3.63E-08 |
| rs109694327 | 10 | 67248322 | A | G | 743 | 5.508 | 3.63E-08 |
| rs136661248 | 10 | 67231936 | G | T | 743 | 5.508 | 3.63E-08 |
| rs110008873 | 10 | 67276588 | A | G | 743 | 5.508 | 3.63E-08 |
| BovineHD1100001194 | 11 | 3312404 | T | C | 280 | 3.633 | 0.00028 |
| rs134083327 | 12 | 10040464 | G | A | 2406 | 6.764 | 1.34E-11 |
| rs134046542 | 12 | 54289535 | G | A | 2406 | 6.728 | 1.72E-11 |
| rs110121749 | 12 | 46243629 | A | C | 2406 | 6.627 | 3.43E-11 |
| 13:81673732 | 13 | 81673732 | C | T | 150 | 5.483 | 4.17E-08 |

*(Continued)*

**Table 3.** (Continued)

| Marker name | CHR | Genomic position | Allele 1 | Allele 2 | Weight* | Z-score | P-value |
|---|---|---|---|---|---|---|---|
| ARS-BFGL-NGS-109467 | 13 | 28331553 | A | G | 280 | 3.76 | 0.00017 |
| BovineHD1300013919 | 13 | 47642260 | A | C | 280 | 3.624 | 0.00029 |
| BovineHD1400017260 | 14 | 62204044 | T | C | 280 | 4.07 | 4.70E-05 |
| rs133977719 | 14 | 28398738 | T | C | 802 | 3.737 | 0.000186 |
| BovineHD1500006429 | 15 | 24508531 | A | C | 150 | 5.598 | 2.17E-08 |
| 15:25797132 | 15 | 25797132 | A | G | 150 | 5.451 | 5.00E-08 |
| rs42691659 | 16 | 63144583 | C | T | 2406 | 6.445 | 1.15E-10 |
| rs110766243 | 17 | 32120758 | G | A | 2406 | 6.77 | 1.29E-11 |
| rs134209520 | 17 | 48682081 | T | C | 743 | 5.264 | 1.41E-07 |
| rs109484372 | 17 | 44884744 | A | G | 743 | 5.229 | 1.70E-07 |
| BovineHD1700000175 | 17 | 713923 | A | G | 150 | 5.223 | 1.76E-07 |
| Hapmap48751-BTA-41232 | 17 | 58016927 | A | G | 280 | 3.775 | 0.00016 |
| BovineHD1700016427 | 17 | 58018395 | A | C | 280 | 3.673 | 0.00024 |
| BovineHD1800009917 | 18 | 32762956 | A | G | 150 | 4.77 | 1.84E-06 |
| rs134127572 | 19 | 36424186 | A | G | 2406 | 6.958 | 3.45E-12 |
| 19:24494923 | 19 | 24494923 | G | C | 150 | 5.851 | 4.89E-09 |
| rs110820800 | 19 | 59382179 | C | A | 802 | 3.748 | 0.000178 |
| rs109366906 | 20 | 18472746 | T | C | 2406 | 6.558 | 5.46E-11 |
| ARS-BFGL-NGS-87102 | 20 | 49859323 | A | G | 150 | 5.404 | 6.52E-08 |
| rs110957960 | 20 | 20494060 | A | G | 1604 | 5.26 | 1.44E-07 |
| rs136158794 | 23 | 39522823 | C | T | 2406 | 6.998 | 2.60E-12 |
| rs133044483 | 23 | 39502283 | C | A | 2406 | 6.998 | 2.60E-12 |
| rs110389869 | 24 | 54044457 | A | G | 2406 | 6.748 | 1.50E-11 |
| rs135137105 | 24 | 22424171 | T | C | 2406 | 6.449 | 1.13E-10 |
| 28:21771233 | 28 | 21771233 | C | T | 150 | 5.478 | 4.30E-08 |
| BTB00987935 | 28 | 35294673 | C | T | 287 | 3.658 | 0.000254 |
| ARS-BFGL-NGS-12759 | 28 | 15739975 | A | G | 280 | 3.624 | 0.00029 |

CHR: Chromosome.

* Weight indicates the total sample size contributing to the meta-analysis for each SNP, corresponding to the sample-size–weighted Z-score approach implemented in the METAL program.

Utilizing the CytoHubba plugin in Cytoscape [49], five genes: *TRPV3*, *TRPV1*, *ANK3*, *PKD2*, and *SHPK* were identified as central hub genes within the interaction network (Fig 3).

In general, gene expression networks are constructed in Cytoscape based on the degree of correlation or interaction among relevant genes [50]. The ClueGO and CluePedia plugins in Cytoscape were employed to identify and visualize the biological, molecular, and cellular functions of genes derived from the GWAS meta-analysis. The result provides an integrated visualization of three major GO categories (Fig 4a–4c).

Each node in the network represents a GO term, and the edges indicate functional relationships based on shared gene involvement and dependency. The candidate genes were found to be involved in a wide range of biological and molecular functions, including metabolic and enzymatic pathways to processes associated with morphogenesis, immune response, and cellular structure organization. This functional network does not only point out the different functions of these genes in various biological contexts but also highlights pathways which can be used as future targets of functional validation or mechanistic studies which can be applied in the trait of interest. Under BP term, various important pathways

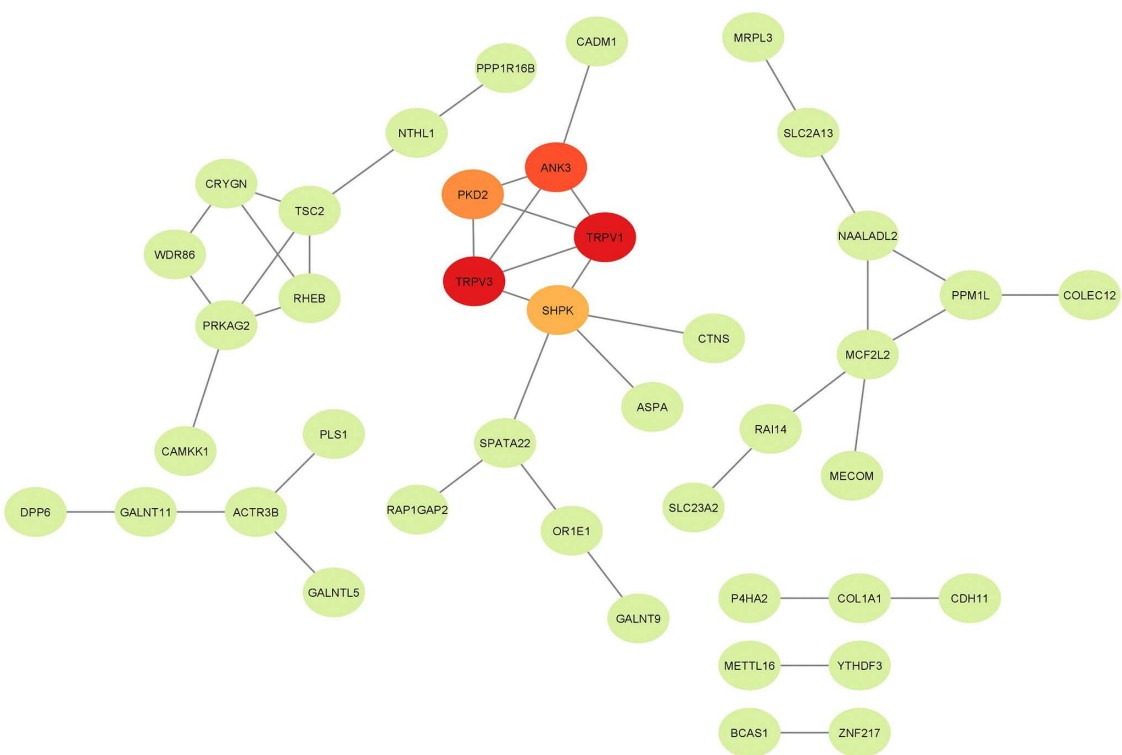

**Fig 3. PPI network of candidate genes and hub genes.** The network illustrates the interactions among candidate genes, with node color intensity indicating their degree of connectivity. The top five hub genes: *TRPV3*, *TRPV1*, *ANK3*, *PKD2*, and *SHPK,* are shown in red and orange, representing higher centrality within the network.

were identified. Among them, there are the acetate metabolic process, which entails the *ASPA* gene, and the cholesterol biosynthetic process via 24,25-dihydrolanosterol, regulated by *CYP51A1*, which underscores the gene's role in cholesterol metabolism. Immune-related processes, such as susceptibility to natural killer cell-mediated cytotoxicity and cytoskeletal pathways like actin cortical patch organization, further illustrate the involvement of candidate genes in immune modulation and cellular architecture. In the MF category, the network was predominantly enriched for pathways related to enzymatic activity and membrane transport. For instance, snRNA (adenine-N6)-methylation activity associated with *METTL16* and betaine-aldehyde dehydrogenase (NAD+) activity linked to *ALDH7A1* highlight functional roles in RNA methylation and osmolyte metabolism, respectively. Within the CC category, genes were associated with structures central to RNA processing and extracellular matrix organization. The post-spliceosomal complex, represented by *PRPF18*, is essential for post-transcriptional RNA splicing, while the collagen type I trimer cluster, including *COL1A1*, reflects the contribution of candidate genes to extracellular matrix integrity and tissue stability.

To elucidate the functional roles of the identified genes, an enrichment analysis using the BiNGO plugin of Cytoscape was performed, with regard to biological processes and molecular functions. The biological process analysis revealed significant enrichment of pathways associated with cytolysis, cell wall organization or biogenesis, cell wall macromolecule metabolism, and catabolism (Fig 5a). These findings indicate that the selected genes are actively involved in essential pathways related to the construction, degradation, and remodeling of cell walls. Also, their participation in cytolytic processes indicates their potential roles in cell death mechanisms that result in the disruption of cellular integrity and release of intracellular contents. All of these pathways may be critical for regulating diverse cellular activities, including stress

Fig 4. Functional enrichment analysis of candidate genes using ClueGO and CluePedia in Cytoscape. (a) Gene ontology enrichment for Biological Processes (BP); (b) Cellular Components (CC); (c) Molecular Functions (MF).

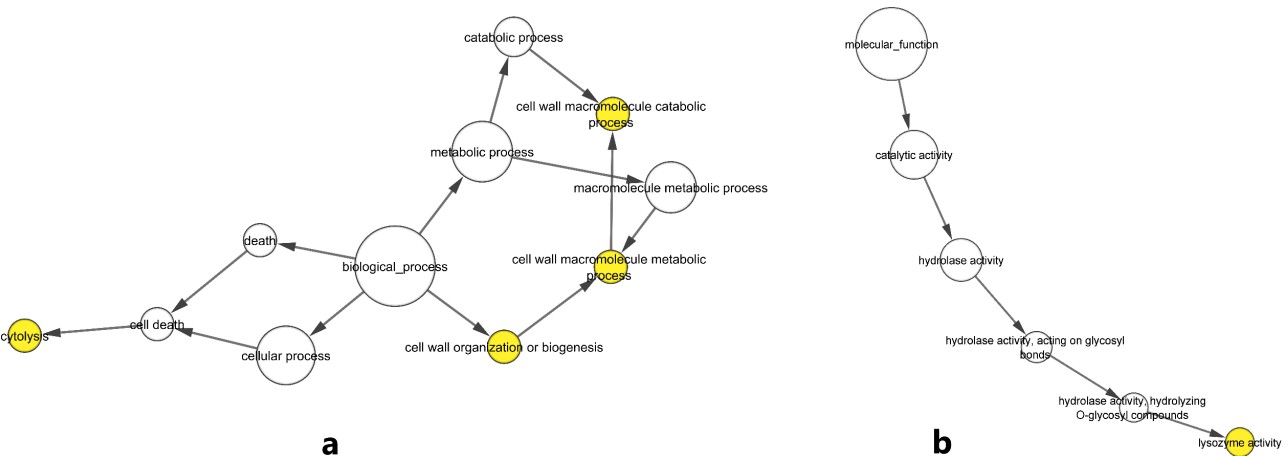

Fig 5. GO analysis of candidate genes using BiNGO. (a) Gene ontology enrichment for biological processes; (b) Gene ontology enrichment for molecular functions. Node color intensity reflects the significance of GO term enrichment (adjusted P-value), with darker colors indicating more significant enrichment.

responses, immune reactions, and tissue regeneration processes. In terms of molecular functions, our analysis identified significant enrichment in enzymatic activities, notably lysozyme activity (Fig 5b).

Lysozymes are enzymes that are part of the regulation of microbes and cellular protection against pathogenic invasion by catalyzing the hydrolysis of glycosidic bonds in the polysaccharide frameworks especially bacterial cell walls [51]. The identified activity of lysozyme is hierarchically organized in terms of cascading of molecular functions that include catalytic

activity, hydrolase activity, hydrolase activity targeting glycosyl bonds, and eventually, hydrolase activity targeting O glycosyl compounds. These findings underscore the targeted catalytic and hydrolytic activity of the candidate genes in the various biological and cellular pathways.

In the present study, the GeneMANIA was utilized to investigate the interactions among candidate genes at both the gene and protein levels. In this analysis, the candidate genes that were identified by genome-wide association meta-analysis were evaluated within the context of an integrated interaction network (Fig 6).

The majority of interactions observed within the constructed network were classified as physical interactions, comprising 70.9% of all connections. These interactions reflect direct binding events between proteins encoded by the candidate genes, suggesting their participation in shared protein complexes or molecular pathways at the protein level. The high prevalence of such interactions implies potential synergistic functions among these gene products within the cellular environment. Co-expression interactions represented 16.01% of the network. These associations are inferred from correlated gene expression patterns across various biological contexts and conditions. High levels of co-expression typically indicate

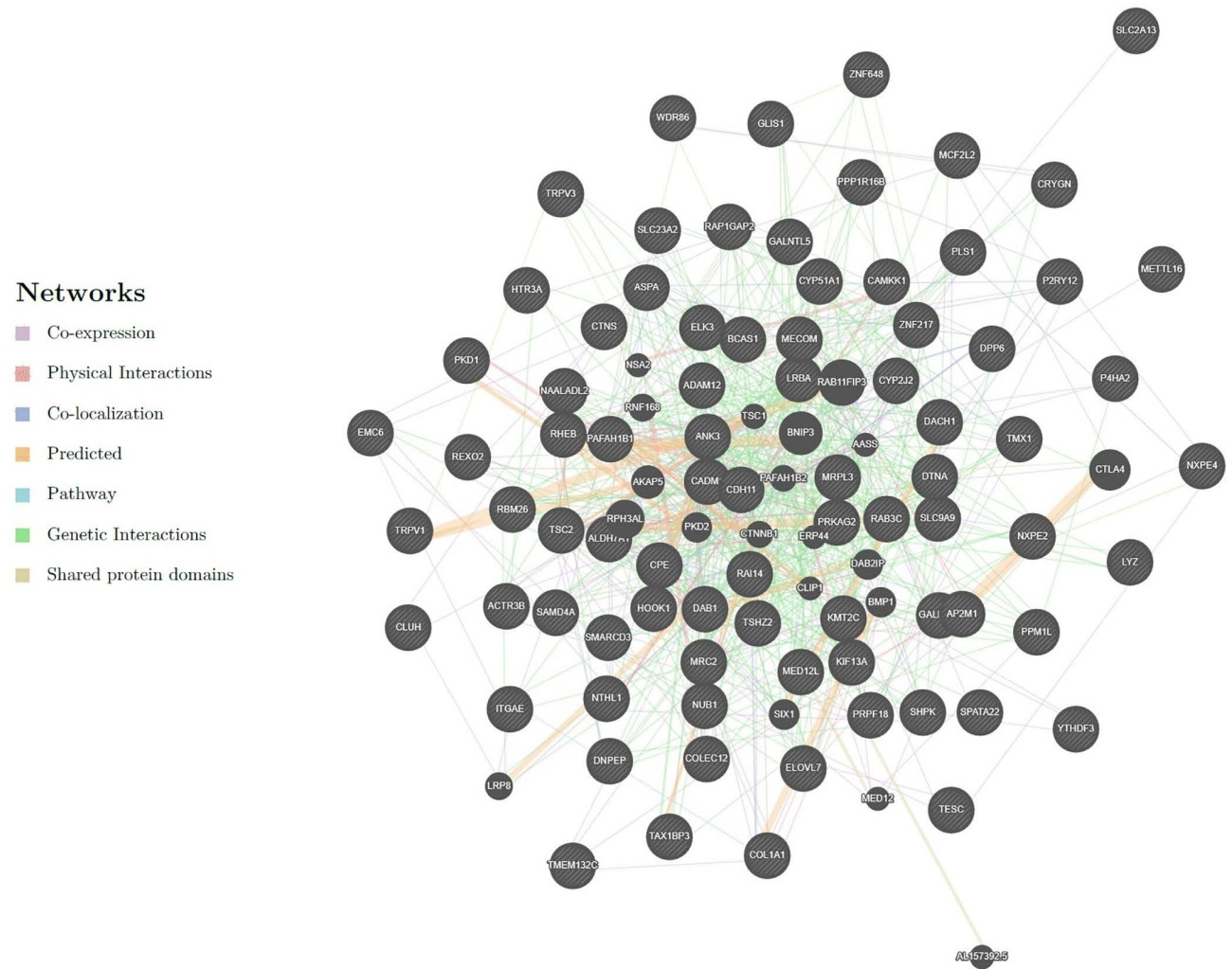

**Fig 6. Gene interaction network of candidate genes generated by GeneMANIA.** The network integrates co-expression, physical interactions, pathways, co-localization, and other functional links among candidate genes. Edge colors indicate interaction types as defined in the legend.

coordinated gene regulation in response to shared environmental stimuli or involvement in common biological pathways. Predicted interactions accounted for 4.96% of total connections. Although these interactions have not yet been experimentally validated, they are derived from computational modeling and suggest potential functional relationships. Their presence, alongside confirmed physical and co-expression interactions, may point to genes with previously unrecognized but functionally analogous roles, highlighting targets for future experimental validation. Other categories included co-localization (3.22%), genetic interactions (2.63%), pathway-based connections (1.74%), and shared protein domains (0.55%). Detailed information on GeneMANIA interactions, including co-expression gene pairs and associated network data, is provided in S4 Table. A comprehensive functional analysis of the identified genes was performed by using the DAVID web tool. The results were systematically grouped into four primary categories: BP, CC, MF, and KEGG signaling pathways. The detailed outputs are presented in Tables 4 and 5.

On the BP category, the candidate genes were highly enriched in a wide variety of specialized pathways, most notably the defense response to bacteria, a process that also involves *COLEC12* and *LYZ2*, *SLC9A9* and *LYSB*, indicating a key role in innate immunity.

Other enriched pathways included vesicular transport along microtubules, negative regulation of the JNK cascade, regulation of small GTPase-mediated signaling, calcium ion import from the plasma membrane, and regulation of mRNA stability. The candidate genes in the CC category were majorly foundin structures such as the endomembrane system, cytoplasm, basolateral plasma membrane, nuclear speck, cell membrane, and Golgi apparatus. Notably, genes such as *RHEB, TMX1, SLC2A13, GALNTL5*, *CTNS*, and *PKD1* were centrally involved in the endomembrane system. More specialized components, including the fibrillar center, stereocilium, and histone deacetylase complex, were also highlighted, suggesting roles in nuclear activity and the regulation of gene expression. At the MF category, enriched terms included calcium channel activity, involving *TRPV3, TRPV1*, and *PKD1*, as well as beta-catenin binding, protein binding, and more specialized molecular activities such as single-atom ion channel activity, galactosaminyltransferase activity, metal ion binding, enzyme inhibition, lysozyme activity, and phosphorylated protein binding. These results point to the broad involvement of the gene set in ion transport, epigenetic regulation, RNA processing, and intracellular signaling, functions that may underpin tissue-specific physiological roles. KEGG pathway analysis further revealed significant enrichment in mucin-type O-glycan biosynthesis, other types of O-glycan biosynthesis, lysine degradation, metabolic pathways, and signaling routes such as the longevity regulating pathway, regulation of TRP channels by inflammatory mediators, histidine metabolism, and thyroid hormone signaling. Among these, the mucin type O-glycan biosynthesis pathway ranked highest in priority, with the lowest p-value (0.0102) and a high enrichment score (19.07). *GALNT11, GALNTL5,* and *GALNT9* were involved in a variety of overlapping pathways indicating a central role in glycoprotein synthesis and intracellular interaction.

## Meta-analysis of genomic and pedigree-based heritability estimates

The effect sizes and heterogeneity statistics for genomic and pedigree-based heritability estimates of methane-related traits in cattle obtained from the random-effects meta-analysis model are presented in Tables 6 and 7, separated by the approach used (genomic and pedigree).

In the genomic analyses, METP showed the highest heritability estimate (0.296), indicating a moderate to high genetic contribution to phenotypic variance. On the other hand, the genomic estimate of METC was significantly lower (0.124). There was significant heterogeneity between the studies of genomic METP, whereas METC showed very low heterogeneity.

The most heritable trait according to pedigree-based heritability estimates was METP (0.299). Methane yield (METY) also presented a similarly moderate heritability estimate (0.293). Moderate heritability estimates were observed for methane intensity (METINT; 0.257) and methane concentration (METC; 0.193). The lowest heritability estimate among the evaluated traits was recorded for residual methane (RMET; 0.167).

**Table 4. Gene ontology analysis performed by the DAVID web tool.**

| BP | Term | P-value | Genes | Fold enrichment |
|---|---|---|---|---|
| | GO:0042742~defense response to bacterium | 0.027767 | COLEC12, LYZ2, SLC9A9, LYSB | 6.044201 |
| | GO:0047496~vesicle transport along microtubule | 0.063015 | KIF13A, PAFAH1B1 | 30.39878 |
| | GO:0046329~negative regulation of JNK cascade | 0.063015 | MECOM, PAFAH1B1 | 30.39878 |
| | GO:0051056~regulation of small GTPase-mediated signal transduction | 0.084307 | RAP1GAP2, TSC2 | 22.46866 |
| | GO:0098703~calcium ion import across plasma membrane | 0.084307 | TRPV3, TRPV1 | 22.46866 |
| | GO:0043488~regulation of mRNA stability | 0.087809 | YTHDF3, SAMD4A | 21.53247 |
| **MF** | GO:0005262~calcium channel activity | 0.023615 | TRPV3, TRPV1, PKD1 | 12.47166 |
| | GO:0008013~beta-catenin binding | 0.044274 | CDH11, TAX1 BP3, MED12L | 8.850859 |
| | GO:0005515~protein binding | 0.051124 | PPP1R16B, CLUH, SAMD4A, TRPV3, WDR86, ANK3, TRPV1, PKD1, COL1A1, NUB1, MECOM, P4HA2, TAX1 BP3, RAI14 | 1.736561 |
| | GO:0005261~monoatomic cation channel activity | 0.06957 | TRPV3, PKD1 | 27.43766 |
| | GO:0004653~polypeptide N-acetylgalactosaminyltransferase activity | 0.07292 | GALNT11, GALNT9 | 26.13111 |
| | GO:0046872~metal ion binding | 0.079312 | COLEC12, RBM26, PPM1L, KMT2C, TSHZ2, GLIS1, COL1A1, P2RX5, RHEB, NTHL1, ADAM12, ZNF217, ASPA | 1.668333 |
| | GO:0005230~extracellular ligand-gated mono-atomic ion channel activity | 0.079585 | HTR3A, TRPV1 | 23.85884 |
| | GO:0004857~enzyme inhibitor activity | 0.079585 | PPP1R16B, TMX1 | 23.85884 |
| | GO:0003796~lysozyme activity | 0.082899 | LYZ2, LYSB | 22.86472 |
| | GO:0001227~DNA-binding transcription repressor activity, RNA polymerase II-specific | 0.087497 | DACH1, ZNF217, GLIS1, ELK3 | 3.7715 |
| | GO:0051219~phosphoprotein binding | 0.089493 | TRPV1, PAFAH1B1 | 21.10589 |
| **CC** | GO:0012505~endomembrane system | 0.002392 | RHEB, TMX1, SLC2A13, GALNTL5, CTNS, PKD1 | 6.390518 |
| | GO:0005737~cytoplasm | 0.026704 | SLC23A2, PRKAG2, CAMKK1, DNPEP, KIF13A, METTL16, MCF2L2, PLS1, RAP1GAP2, PPP1R16B, CLUH, DTNA, YTHDF3, TESC, SAMD4A, TSC2, ANK3, REXO2, CYP2J30, COL1A1, DAB1, RHEB, HOOK1, CDH11, TAX1 BP3, ASPA | 1.495213 |
| | GO:0016323~basolateral plasma membrane | 0.032552 | SLC23A2, TAX1 BP3, ANK3, PKD1 | 5.687561 |
| | GO:0016607~nuclear speck | 0.038041 | PPP1R16B, PRPF18, DACH1, MECOM, ZNF217 | 3.906292 |
| | GO:0016020~membrane | 0.048595 | P2RY12, COLEC12, RAB3C, SLC23A2, TMEM132C, DTNA, TMEM178B, CADM1, PPM1L, LRBA, CYP51A1, NXPE2, CTNS, ANK3, NAALADL2, MRC2, TMEM233, DPP6, RHEB, ADAM12, GALNTL5, CPE, EMC6, GALNT9 | 1.448753 |
| | GO:0005829~cytosol | 0.055484 | RAP1GAP2, PPM1L, LRBA, KMT2C, TESC, SAMD4A, PRKAG2, NUB1, DNPEP, DACH1, RHEB, P4HA2, SHPK, ALDH7A1, ASPA, RAI14, PAFAH1B1, PLS1 | 1.564427 |
| | GO:0005794~Golgi apparatus | 0.064664 | GALNT11, DACH1, GALNTL5, CPE, TSC2, PKD1, GALNT9 | 2.436532 |
| | GO:0001650~fibrillar center | 0.074174 | SAMD4A, TAX1 BP3, RAI14 | 6.613443 |
| | GO:0032420~stereocilium | 0.083355 | PAFAH1B1, PLS1 | 22.75024 |
| | GO:0000118~histone deacetylase complex | 0.099183 | MECOM, ZNF217 | 18.95854 |

BP: Biological Processes; CC: Cellular Components; MF: Molecular Functions.

**Table 5. Enriched functional pathways.**

| Term | P-value | Genes | Fold enrichment |
|------|---------|-------|-----------------|
| bta00512:Mucin type O-glycan biosynthesis | 0.010210213 | GALNT11, GALNTL5, GALNT9 | 19.0744186 |
| bta00514:Other types of O-glycan biosynthesis | 0.017229165 | GALNT11, GALNTL5, GALNT9 | 14.51314459 |
| bta00310:Lysine degradation | 0.03379633 | MECOM, KMT2C, ALDH7A1 | 10.11522199 |
| bta01100:Metabolic pathways | 0.046363087 | GALNT11, KMT2C, CYP51A1, ELOVL7, CYP2J30, MECOM, P4HA2, SHPK, GALNTL5, ALDH7A1, ASPA, GALNT9, PAFAH1B1 | 1.756498779 |
| bta04211:Longevity regulating pathway | 0.05909808 | RHEB, PRKAG2, TSC2 | 7.417829457 |
| bta04750:Inflammatory mediator regulation of TRP channels | 0.083790118 | TRPV3, TRPV1, CYP2J30 | 6.069133192 |
| bta00340:Histidine metabolism | 0.092335928 | ALDH7A1, ASPA | 20.23044397 |

**Table 6. Effect size and heterogeneity of the genomic heritability estimates for methane-related traits based on the random-effects meta-analysis model.**

| Trait | Study | N | $h^2$ | SE | 95% CI | P-value | Q | P-value | $I^2$ |
|-------|-------|---|-------|-----|--------|---------|---|---------|-------|
| METP | 14 | 19 | 0.296 | 0.035 | 0.227-0.366 | 0.000 | 164.282 | 0.000 | 89.043 |
| METC | 5 | 5 | 0.124 | 0.009 | 0.107-0.141 | 0.000 | 2.077 | 0.000 | 0.000 |

METP: methane production; METC: methane concentration; N: Number of literature estimates; SE: Standard error; 95% CI: 95% confidence interval.

**Table 7. Effect size and heterogeneity of the pedigree-based heritability estimates for methane-related traits based on the random-effects meta-analysis model.**

| Trait | Study | N | $h^2$ | SE | 95% CI | P-value | Q | P-value | $I^2$ |
|-------|-------|---|-------|-----|--------|---------|---|---------|-------|
| METP | 28 | 38 | 0.303 | 0.020 | 0.264-0.342 | 0.000 | 705.149 | 0.000 | 94.753 |
| METY | 12 | 18 | 0.293 | 0.030 | 0.234-0.353 | 0.000 | 159.702 | 0.000 | 89.355 |
| METINT | 11 | 14 | 0.257 | 0.038 | 0.183-0.330 | 0.000 | 243.987 | 0.000 | 94.672 |
| METC | 7 | 12 | 0.207 | 0.022 | 0.164-0.250 | 0.000 | 58.583 | 0.000 | 81.223 |
| RMET | 4 | 10 | 0.167 | 0.014 | 0.138-0.195 | 0.000 | 4.262 | 0.000 | 0.000 |

METP: methane production; METY: methane yield; METINT: methane intensity; METC: methane concentration; RMET: residual methane; N: Number of literature estimates; SE: Standard error; 95% CI: 95% confidence interval.

The pedigree-based estimates also had a significant heterogeneity. There was the highest heterogeneity in METP (P<0.001). Other traits, excluding RMET, displayed substantial heterogeneity.

The forest plots of individual studies and the overall genomic and pedigree-based heritability estimates for METP are shown in Figs 7 and 8, respectively. The heritability estimates for other traits are presented in S1–S5 Figs.

To determine the possible publication bias of the meta-analytic heritability of the traits associated with methane, the regression test of Egger was used in all the analyzed traits, and the results are described in Table 8.

In the case of genomic METP, a statistically significant publication bias was found by Egger's test. Therefore, Duval and Tweedie's trim-and-fill method revealed two studies that might be missing on the left side of the funnel plot. The inclusion of these hypothetical studies produced the adjusted estimate of heritability of 0.273, which suggests that the original estimate may be inflated by publication bias. The funnel plot illustrating this analysis is presented in Fig 9.

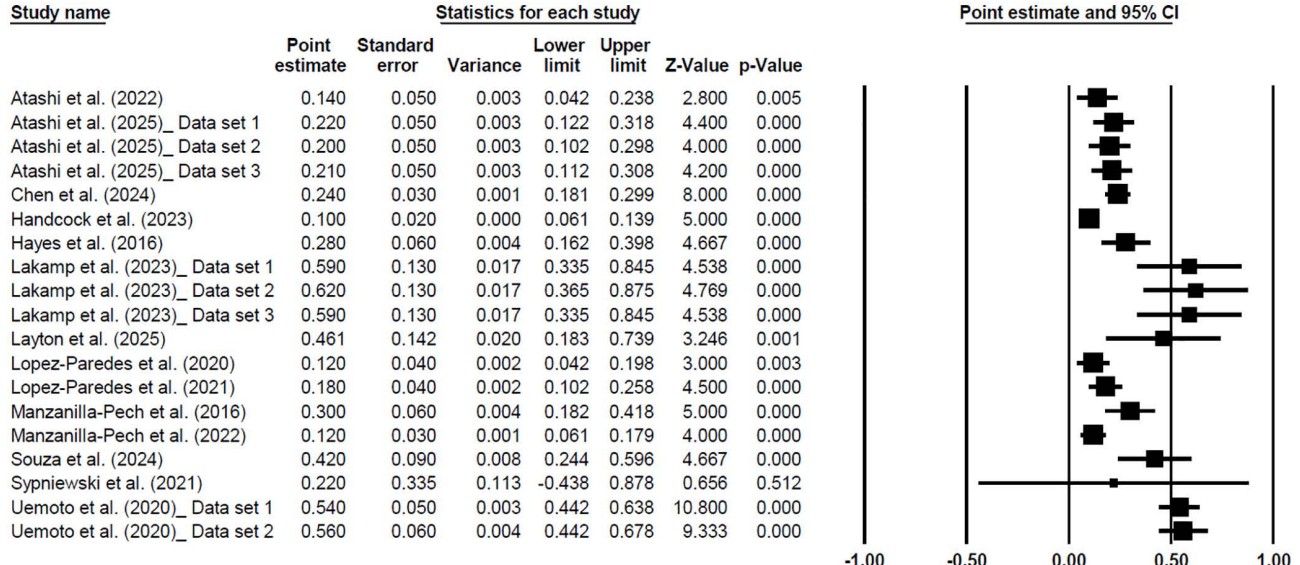

**Fig 7. The forest plot of individual studies and the overall outcome for genomic heritability estimates of Methane Production.** The mean effect size, calculated according to a random effects model, is indicated by the diamond at the bottom of each plot. The size of the squares illustrates the weight of each study relative to the mean effect size. Smaller squares represent less weight. The horizontal bars represent the 95% CI for the study.

Genomic METC did not show any significant publication biases (P > 0.10). Although the trim-and-fill method indicated the existence of two missing studies. After inclusion of these hypothetical studies, the adjusted heritability estimate was 0.114. The funnel plot for this trait is provided in S6 Fig.

In the pedigree-based analyses, Egger's test didn't show any significant publication bias for METP, METY, METINT, and METC (P > 0.10). Based on this, the trim-and-fill method did not identify any missing studies, and the adjusted heritability were consistent with the initial values. However, for RMET, significant publication bias was detected (P = 0.00041), with four missing studies estimated on the left side of the funnel plot. The adjusted heritability for RMET decreased to 0.154, indicating a meaningful reduction from the original estimate. Fig 10 shows the funnel plot of pedigree-based METP and similar plots of other traits are shown in S7–S10 Figs.

## Discussion

The investigation of methane-related traits across different cattle breeds represents a fundamental component of genomic strategies aimed atmitigating greenhouse gas emissions in the livestock production industry [3,52]. Genetic selection for reducing methane production is considered a sustainable approach because the realized genetic gainsare cumulatively transmitted to subsequent generations, thereby providing long-term environmental benefits [53,54]. In this context, the present research, employed a rigorous and comprehensive meta-analysis of GWAS to integrate heterogeneous evidence on methane traits across multiple breeds and experimental designs. Through systematic screening and inclusion of eligible studies, a consolidated dataset encompassing 4,336 animals was compiled, substantially increasing the statistical power compared with individual studies.

The Meta-analytical GWAS identified 173 SNPs associated with methane-related traits, of which 74 remained statistically significant after Bonferroni correction (P < 2.90 × 10⁻⁴). These significant SNPs were distributed across various chromosomes, underscoring the polygenic nature underlying methane production traits in cattle [20]. Further investigation of adjacent genes and descriptive studies showed that no common significant SNPs were found across studies, which

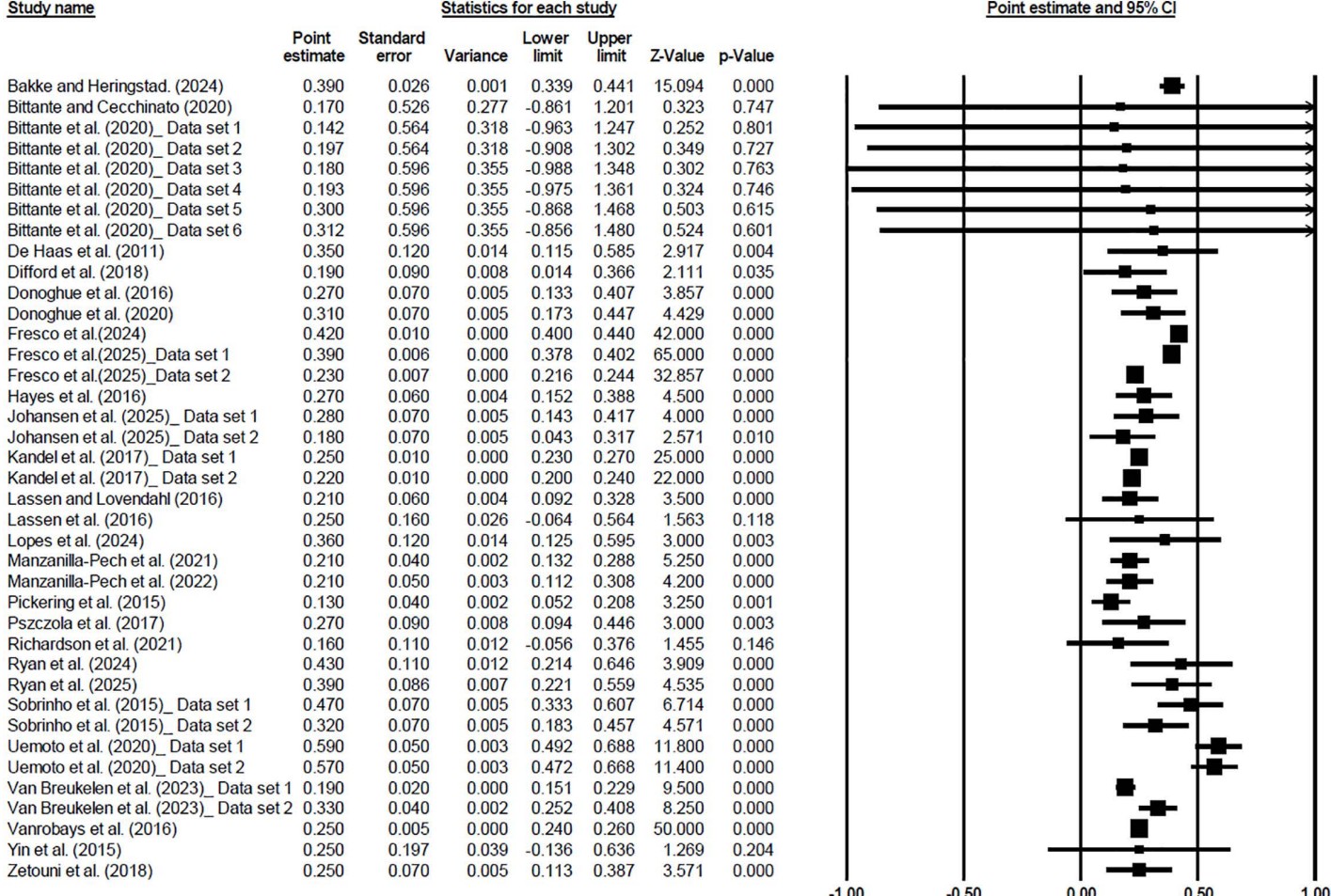

**Fig 8. The forest plot of individual studies and the overall outcome for pedigree-based heritability estimates of Methane Production.** Details are provided in Fig.7.

**Table 8. Summary of publication bias diagnostics and funnel plot asymmetry correction using the trim-and-fill method for traits with no detected heterogeneity. The table presents results of statistical tests for publication bias and adjusted mean heritability estimates following the imputation of potentially missing studies.**

| Trait* | P-value of Egger's Test | Trim and Fill | | |
|---|---|---|---|---|
| | | Missing | Mean | 95% CI |
| METP-G | 0.00166 | 2 | 0.272 | 0.204-0.340 |
| METC-G | 0.21440 | 2 | 0.120 | 0.103-0.136 |
| METP-P | 0.97772 | 0 | 0.303 | 0.264-0.342 |
| METY-P | 0.26428 | 0 | 0.293 | 0.234-0.353 |
| METINT-P | 0.46266 | 0 | 0.257 | 0.183-0.330 |
| METC-P | 0.76476 | 0 | 0.207 | 0.164-0.250 |
| RMET-P | 0.00081 | 4 | 0.154 | 0.129-0.180 |

*For traits, see Table 7. G: Genomic heritability; P: Pedigree-based heritability.

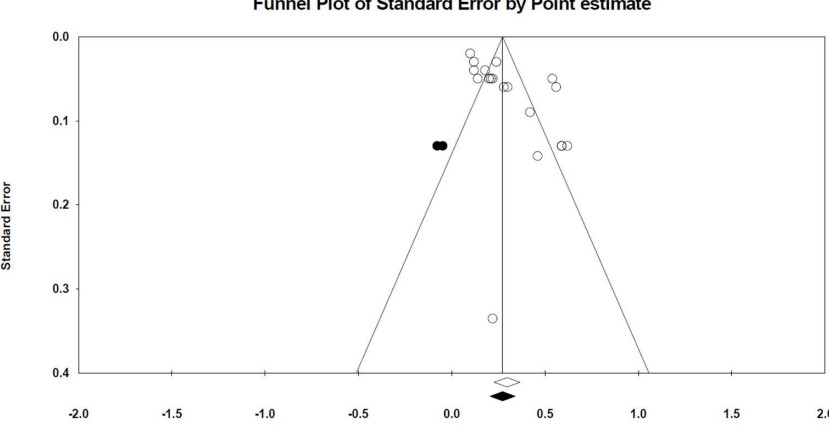

**Fig 9. The funnel plot of the genomic heritability estimate for METP.** Solid dots indicate the potentially missing studies identified through the trim-and-fill method. Open diamonds represent the pooled estimate and 95% confidence interval (CI) based solely on the published studies, whereas solid diamonds reflect the adjusted pooled estimate and CI after accounting for the imputed studies.

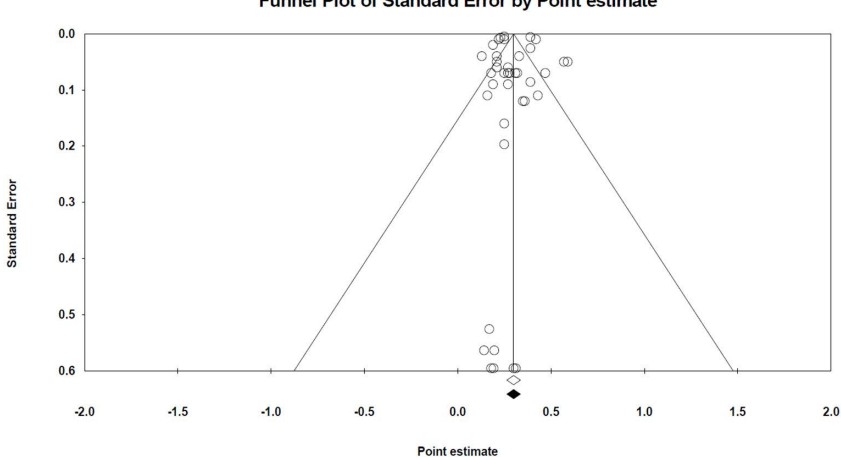

**Fig 10. The funnel plot of the pedigree-based heritability estimate for METP.** Details are provided in Fig 9.

was probably due to heterogeneity in study designs, breed composition, and statistical methodologies. The meta-analysis methodology used in the study helped combine and amplify statistical power of independent studies, thus, providing a wider range of expected genomic areas of methane traits. In addition to the discovery of the markers, the research provides a synthesis of the estimates of the genetic parameters, which clarifies the heritability of methane traits, using the evidence of genomic and pedigree [23,29,30]. When an extra quantitative meta-analysis based on the CMA framework was included, the estimates of heritabilities and heterogeneity parameters were refined to show a consistent picture of moderate genetic determination of production-level traits. These results align with previous reports in Holstein and Danish Holstein populations, where heritability estimates for methane production (0.16–0.49) confirm the feasibility of achieving genetic progress through selection [52,53].

Furthermore, the convergence between methane traits and correlated indicators such as feed efficiency underscores the practical breeding relevance of these results. Traits with positive genetic correlations with methane production can be efficiently integrated into selection indices, and thus, goals will be reached in both environmental sustainability and efficiency of production [19,54,55]. Taken together, this meta-analysis would not only summarize the existing evidence of the heterogeneous studies, but also point out the new possibilities of genomic optimization of the methane traits, which would provide a statistically reinforced and biologically oherent framework for sustainable livestock breeding.

The present dual analytical approach combining meta-GWAS with the independent meta-analysis of heritability estimates provides a full framework of the evaluation of both the genomic signals and the quantitative genetic variation of methane emission traits in cattle. This method does not determine the entire genetic architecture, but rather defines the recurrent patterns found across studies and estimates the strength of additive genetic control, which can be used as a biologically and statistically supported basis of future validation. The findings underscore the fact that the methane characteristics, though polygenic and environmentally adjusted, are moderately heritable and hence they should be considered as the possible targets of selection in breeding programs. Notably, this synthesis fills the gap between molecular level understanding of SNP and pathway analysis and population level demonstrations of heritable variation which contributes to the better understanding of genomic data into practical breeding. The study thus leads to a more integrative view of the possibility of host genetics to affect methane phenotyping so that they may serve as the basis of evidence in the design of future genomic prediction and selection systems. The production of methane-associated characteristics should be targeted to achieve global climate and sustainability outcomes and preserve the efficiency of production. These traits are economically feasible and genetic gains can be increased during consecutive generations. In light of global initiatives aiming to reduce methane to lower the amount of methane released into the atmosphere by the year 2050, the use of genetic enhancement of cattle with special attention to methane characteristics is a factual and realistic avenue that can be used to meet climate goals. However, further merging of functional genomics and metagenomic data will be necessary in order to transform these statistical linkages into informative and practical breeding tools [53,56].

To further investigate the biological context of the identified genes from SNPs, post-GWAS enrichment analyses including GO terms and KEGG pathways were performed. *TRPV1 and TRPV3* were also discovered as hub genes in the protein-protein interaction network and were enriched in significant biological pathways, such as calcium ion import across the plasma membrane (GO:0098703), calcium channel activity (GO:0005262), and the regulation of inflammatory mediators by TRP channels (bta04750). The convergent pattern of these overlapping enrichments in independent analyses supports the idea that the epithelial ion-transport processes are the emerging biological signal in representations of methane-related traits. These genes have been identified earlier to be related with SNP predictive of methane emission in GWAS on Holstein cattle [19].

This process can aid the metabolism of nitrogen and can possibly affect the uptake of ammonia which has been indirectly associated with the emission of nitrous oxide into the environment. Although the present meta-analysis does not provide direct functional evidence linking *TRPV3* to methane emission, its enrichment in ion transport and signaling pathways, together with prior expression studies in the rumen, indicates a putative role in modulating epithelial bioelectrical properties that could influence rumen fermentation dynamics and microbial composition. Hence, *TRPV3* may act as a contextually relevant candidate gene having an indirect effect on the emission of methanethrough nitrogen and microbial homeostasis, though this remains to be experimentally validated [57,58].

*TRPV1* was enriched in the same GO terms and the same KEGG pathway (e.g., GO:0098703, GO:0005515 and bta4750) as *TRPV3*, but its expression has a more peripheral distribution. It is primarily located in systemic calcium signaling and it is not specifically expressed in the rumen epithelium. Therefore, its identification in GWAS may reflect broader physiological correlations or linkage disequilibrium with functionally relevant variants rather than direct causality within the rumen.

In addition to the *TRP* family, several genes were enriched in metabolic and structural pathways, illustrating the multi-factorial nature of methane-related traits. *SHPK*, encoding sedoheptulokinase, catalyzes the conversion of sedoheptulose to sedoheptulose-7-phosphate in the non-oxidative phase of the pentose phosphate pathway (PPP). This pathway plays an essential role in redox balance and NADPH production, supporting lipid and nucleotide biosynthesis [59]. *SHPK* was localized to the cytosol (GO:0005829) and enriched in metabolic pathways (bta01100), suggesting its function in maintaining cellular energy and redox homeostasis. Although not directly implicated in methane metabolism, *SHPK's* regulatory function in carbon flux and oxidative balance may indirectly affect feed efficiency, a trait inversely related to methane yield [60].

Also, GO annotations indicate that *ANK3* is involved in several key cellular functions: protein binding (GO:0005515), cytoplasmic localization (GO:0005737), association with the membrane (GO:0016020), and specific targeting to the basolateral plasma membrane in epithelial cells (GO:0016323). Its expression is especially high in excitable and polarized tissues such as brain, muscle, and cardiac cells [61,62], with multiple isoforms generated via alternative splicing [63]. *ANK3* encodes ankyrin-G, a scaffolding protein that anchors membrane proteins to the spectrin–actin cytoskeleton across various cell types [64–66] and consists of multiple ankyrin repeats, a spectrin-binding region, and a regulatory C-terminal domain [67]. While *ANK3* contributes to cell polarity and membrane organization, there is currently no experimental evidence linking it directly to methane metabolism. Its identification across multiple GO categories likely reflects a structural or regulatory cellular role rather than trait-specific functionality [68–70].

Among the genes located near significant SNPs identified in the current meta-analysis (rs43508672, rs43508661, rs43508667, rs29023390, rs43508669), *P4HA2* was notable for its in protein binding (GO:0005515), cytosolic localization (GO:0005829), and metabolic pathways (bta01100). *P4HA2* encodes prolyl 4-hydroxylase subunit alpha-2, an enzyme involved in collagen biosynthesis and extracellular matrix (ECM) organization. Its co-occurrence with ECM-related genes such as *COL1A1* supports the hypothesis that structural remodeling processes may influence tissue metabolism and, indirectly, methane-related physiology [16]. *P4HA2* has also been implicated in metabolic adaptation under hypoxia. It was identified as a hub gene in Tibetan sheep and goats, where it appears to regulate meat quality, skeletal muscle development, and high-altitude hypoxia response [71,72]. In Tibetan sheep, it resides within a breed-specific ROH island associated with hypoxic adaptation [72]. Its expression is also regulated by the p53 and HIF1α signaling axes under hypoxic stress, interacting with pathways such as mTOR and VEGF signaling [73]. Moreover, *P4HA2* has been reported to participate in PI3K/AKT signaling and epithelial–mesenchymal transition (EMT) in non-ruminant models, indicating broader regulatory capacity in cellular proliferation and metabolism [74,75]. The potential implication of the role of *P4HA2* in energy metabolism and ECM remodeling is that it may have indirect effects on methane yield. The ECM influences cellular signaling, energy partitioning, and potentially immune responses, all of which may shape rumen physiology and microbial interactions. The co-localization of ECM-related genes with methane-associated SNPs further underscores their potential regulatory significance, warranting further functional validation in the context of host–microbiome–environment interactions.

Consistent with this ECM-related signal, *COL1A1* was enriched highly in protein binding (GO:0005515), metal ion binding (GO:0046872), cytoplasmic localization (GO:0005737), and collagen-related cluster of ClueGO/CluePedia results. Previous studies reported the downregulation of *COL1A1* in ruminal epithelial cells of high-gain cattle [76], which might be associated with energy metabolism and efficiency of nutrient-transport. *P4HA2* and *COL1A1* both were located within QTL regions associated with methane traits [16], indicating that ECM remodeling may serve as an intermediate layer, which combines cellular metabolism, tissue structure, and energy efficiency, which ultimately determine methane yield. *KIF13A* encodes a kinesin motor protein involved in microtubule-based vesicle transport, contributing to intracellular organization and membrane trafficking. KIF13A contributes to transfers in gastrointestinal epithelia, i.e., rumen where nutrient transport and metabolic regulation is vital in ensuring a stable microbial environment in ruminants [77,78]. Previous transcriptomic analyses identified *KIF13A* as differentially expressed between low- and high-residual feed intake cattle [79], supporting

its potential involvement in energy efficiency pathways. Such mechanisms could indirectly affect methane production through altered nutrient utilization and oxygen availability in the rumen environment [80,81].

The network metabolic regulation was further supported by the enrichment of the *ALDH7A1*, *CYP51A1*, and *SHPK* in metabolic pathways (bta01100), which includes degradation of amino-acids, lipid metabolism, and redox regulation. *ALDH7A1* is involved in the lysine degradation (bta00310) and histidine metabolism (bta00340), which makes it coupled with the feed efficiency and antioxidant defense [82–84]. Similarly, *CYP51A1*, which is associated with sterol/steroid production, is implicated in lipid metabolism and energy use, which are pathways that have been reported to be linked to rumen fermentation and methane variation [20,85]. Taken together these results provide evidence that biological mechanisms of methane characteristics includes ion transport and ECM remodelling, and also energy/redox metabolism, which points to the integrative role of host genomic regulation of methane emission phenotypes.

In general, the post-GWAS analyses confirmed the presence of biological terms that were justified by several enrichment analyses (GO, KEGG, and network), and a interpretation of the results in accordance with the existing data. The results do not publish causality to specific genes but represent functionally consistent networks and pathways that produce a biologically viable pattern of the situation in which cattle variation is linked to methane. These results supplement the quantitative data of heritability meta-analyses and define mechanistic hypotheses which can be validated in empirical studies of functionality and metagenomics in future.

Our meta-analysis distinguished between pedigree-based and genomic heritability estimates for various methane-related traits in cattle (Tables 6 and 7). The integration of both approaches enables a more comprehensive assessment of the genetic potential underlying methane phenotypes while accounting for methodological heterogeneity among studies.

In the present meta-analysis, METP exhibited a moderate heritability (0.296), while METC showed a lower estimate (0.124), suggesting that genetic improvement through selection may be more effective for production-level traits than for concentration-based measures. These findings are in line with previous studies that reported similar or slightly lower genomic heritabilities for methane emissions. For instance, Manzanilla-Pech et al. (2016) estimated heritability values of 0.30 for METP in Angus cattle and 0.23–0.42 for methane-related traits in Holstein populations [86]. Comparable estimates have also been reported in sheep, indicating a consistent genetic basis across ruminants [21,87]. More recent studies applying both direct and indirect phenotyping methods reported genomic heritability estimates in the range of 0.11 to 0.15 [88,89], though lower values may reflect differences in sensor technologies, recording environments, or data structures. These findings support the moderately heritable nature of methane traits. While phenotypic variability may be influenced by measurement approach and animal stage, the consistent presence of additive genetic variation across studies indicates that methane emission traits can be targeted in breeding programs, particularly when supported by accurate genomic evaluations and repeated records for improved reliability [89–92].

The pedigree-based heritability estimates obtained in this meta-analysis further underscore the genetic basis of methane-related traits in cattle. Among the evaluated traits, METP showed the highest heritability, followed closely by methane yield. Moderate heritability was observed for methane intensity, methane concentration, and residual methane, reflecting considerable additive genetic variance across trait definitions. These results are consistent with prior studies that have reported moderate heritabilities for methane traits in both dairy and beef cattle. For example, Kamalanathan et al. (2023) estimated heritabilities of 0.16, 0.27, and 0.21 for METP, METY, and METINT, respectively, in Holstein cattle under pedigree-based models [52]. Similarly, Ryan et al. (2025) observed heritability estimates ranging from 0.15 to 0.39, depending on trait definition and measurement approach [93]. Bittante et al. (2020) also highlighted moderate heritabilities for methane emissions predicted from milk fatty acid profiles, supporting the view that these traits are genetically tractable [94]. The heritability is theoretically non-negative; however, the confidence intervals for some individual studies in forest plots (Figs 7 and 8) were calculated using Wald-type normal approximations (estimate ± 1.96 × SE) within a random-effects meta-analysis framework, which does not impose boundary constraints. Consequently, for estimates with larger standard

errors, the lower bound of the confidence interval may extend below zero due to sampling uncertainty. These negative values do not imply negative heritability but indicate that the true value may be close to zero.

The variation observed across studies is partly attributable to differences in phenotyping strategies, measurement technologies, and environmental conditions. Consistent with these observations, the present meta-analysis reinforces the hypothesis that methane emissions exhibit sufficient genetic variability to be included as selection objectives in breeding programs. Traits such as METP and METY, with moderate heritability, appear particularly promising for genetic improvement. Furthermore, established genetic correlations between methane traits and feed efficiency [95–97], suggest that selection for reduced methane emission may concurrently enhance resource-use efficiency. Integrating METP into breeding indices, alongside careful trait definition and measurement standardization, holds strong potential to support sustainable dairy and beef production systems with a minimal environmental footprint.

## Conclusions

The present study combined two complementary meta-analytical approaches to provide that integration of GWAS and heritability data enabled a comprehensive view of the genetic architecture underlying methane emission traits in cattle. Post-GWAS analyses identified several candidate genes enriched in biological processes that were mainly related to energy metabolism, epithelial ion transport and extra cellular matrix remodelling; these results highlight the multivariate and regulatory aspects of the phenotypes of methane. Taken together, the results advance a more refined understanding of the molecular mechanisms that may indirectly influence methane production, while acknowledging the need for functional validation to establish causality.

In a quantitative genetics perspective, the meta-analysis of heritability revealed consistent moderate genetic heritability, specifically on METP and METY. These findings confirm the fact that methane traits are sufficiently heritable to be targeted within selection criteria, while also defining methodological heterogeneity across studies. Importantly, the consistency of genomic and pedigree-based estimates reinforces the reliability of these parameters and their applicability in breeding programs. Overall, this study highlights the value of combining the evidence of functional genomics with the assessment of breeding in developing breeding strategies that can suppress methane emissions. The results indicate that genomic selection which focuses on traits associated with methane can be practiced and also potentially increase feed efficiency, which can be used to enhance environmentally friendly livestock production. To improve the biological basis of methane mitigation through genetic selection, future studies need to focus on refining phenotyping, expand multi-breed genomic data sets, and experimentally validate candidate genes of critical importance.

## Supporting information

**S1 Checklist. The PRISMA checklist used in this study.**
(DOCX)

**S1 Table. The characteristics of studies included in h$^2$ meta- analysis.**
(DOCX)

**S2 Table. List of the identified genes related to significant SNPs.**
(DOCX)

**S3 Table. Detailed functional annotations for identified genes related to significant SNPs.**
(XLSX)

**S4 Table. Genemania Interactions.**
(DOCX)

**S1 Fig. The forest plot of individual studies and the overall outcome for the genomic heritability estimate of METC.** The mean effect size, calculated according to a random effects model, is indicated by the diamond at the bottom of each plot. The size of the squares illustrates the weight of each study relative to the mean effect size. Smaller squares represent less weight. The horizontal bars represent the 95% CI for the study.
(PDF)

**S2 Fig. The forest plot of individual studies and the overall outcome for the pedigree-based heritability estimate of METC.** Details are provided in S1 Fig.
(PDF)

**S3 Fig. The forest plot of individual studies and the overall outcome for the pedigree-based heritability estimate of METI.** Details are provided in S1 Fig.
(PDF)

**S4 Fig. The forest plot of individual studies and the overall outcome for the pedigree-based heritability estimate of METY.** Details are provided in S1 Fig.
(PDF)

**S5 Fig. The forest plot of individual studies and the overall outcome for the pedigree-based heritability estimate of RMET.** Details are provided in S1 Fig.
(PDF)

**S6 Fig. The funnel plot of the genomic heritability estimate for METC.** Solid dots represent the potentially missing studies that were found using the trim-and-fill method. When theoretically imputed studies are included in the meta-analysis, solid diamonds represent the mean values and CI, and open diamonds represent the mean values and confidence intervals for studies that are currently in the literature.
(PDF)

**S7 Fig. The funnel plot of the pedigree-based heritability estimate for METC.** Details are provided in S6 Fig.
(PDF)

**S8 Fig. The funnel plot of the pedigree-based heritability estimate for METI.** Details are provided in S6 Fig.
(PDF)

**S9 Fig. The funnel plot of the pedigree-based heritability estimate for METY.** Details are provided in S6 Fig.
(PDF)

**S10 Fig. The funnel plot of the pedigree-based heritability estimate for RMET.** Details are provided in S6 Fig.
(PDF)

## Author contributions

**Conceptualization:** Navid Ghavi Hossein-Zadeh.

**Data curation:** Sare Golpasand.

**Formal analysis:** Sare Golpasand.

**Project administration:** Navid Ghavi Hossein-Zadeh.

**Supervision:** Navid Ghavi Hossein-Zadeh.

**Validation:** Navid Ghavi Hossein-Zadeh.

**Writing – original draft:** Sare Golpasand.

**Writing – review & editing:** Navid Ghavi Hossein-Zadeh, Shahrokh Ghovvati.

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
