## [Decision Letter · Decision Letter 0]

5 Feb 2026

Genomic dissection of methane emission traits in cattle: a meta-GWAS and heritability analysis across populations

PLOS One

Dear Dr. Ghavi Hossein-Zadeh,

Thank you for submitting your manuscript to PLOS ONE. After careful consideration, we feel that it has merit but does not fully meet PLOS ONE’s publication criteria as it currently stands. Therefore, we invite you to submit a revised version of the manuscript that addresses the points raised during the review process.

We look forward to receiving your revised manuscript.

Kind regards,

Hayrunnisa Nadaroglu

Academic Editor

PLOS One

Journal Requirements:

1.Please ensure that your manuscript meets PLOS ONE's style requirements, including those for file naming. The PLOS ONE style templates can be found at https://journals.plos.org/plosone/s/file?id=wjVg/PLOSOne_formatting_sample_main_body.pdf and

Reviewers' comments:

Reviewer's Responses to Questions

**Comments to the Author**

1. Is the manuscript technically sound, and do the data support the conclusions?

Reviewer #1: Yes

Reviewer #2: Yes

2. Has the statistical analysis been performed appropriately and rigorously?

Reviewer #1: Yes

Reviewer #2: Yes

3. Have the authors made all data underlying the findings in their manuscript fully available?

Reviewer #1: Yes

Reviewer #2: Yes

4. Is the manuscript presented in an intelligible fashion and written in standard English?

Reviewer #1: No

Reviewer #2: Yes

Reviewer #1: The article contains several grammatical issues (some of which have been highlighted in the text). In addition, self-citations by the authors should be avoided, and alternative references have been suggested.

Palangi, V., & Lackner, M. (2022). Management of enteric methane emissions in ruminants using feed additives: A review. Animals, 12(24), 3452.

Palangi, V., Taghizadeh, A., Abachi, S., & Lackner, M. (2022). Strategies to mitigate enteric methane emissions in ruminants: A review. Sustainability, 14(20), 13229.

Kader Esen, V., Palangi, V., & Esen, S. (2022). Genetic improvement and nutrigenomic management of ruminants to achieve enteric methane mitigation: A review. Methane, 1(4), 342-354.

Reviewer #2: Dear authors,

I have read the manuscript entitled “Genomic dissection of methane emission traits in cattle: a meta-GWAS and heritability analysis across populations” and consider that it is well written and easy to follow. The methodology used is appropriate for the objectives of the study.

I have suggested some minor comments, which can be found in the attached document.

**Do you want your identity to be public for this peer review?** For information about this choice, including consent withdrawal, please see our For information about this choice, including consent withdrawal, please see our Privacy Policy .

Reviewer #1: No

Reviewer #2: **Yes:** Pablo Dominguez-CastañoPablo Dominguez-Castaño

---

## [Author Response · Author response to Decision Letter 1]

6 Feb 2026

With sincere thanks to the respective reviewers for their constructive comments, we are pointing to the changes we made to the revised manuscript. All changes were highlighted within the revised text.

Reviewer #1:

1. Grammar corrections.

AU: Thank you for your suggestions. All corrections were made within the revised text as suggested (Lines 21-22, 30-31, 34-35, 39, 43)

2. The self-citations by the authors should be avoided, and alternative references have been suggested.

Palangi, V., & Lackner, M. (2022). Management of enteric methane emissions in ruminants using feed additives: A review. Animals, 12(24), 3452.

Palangi, V., Taghizadeh, A., Abachi, S., & Lackner, M. (2022). Strategies to mitigate enteric methane emissions in ruminants: A review. Sustainability, 14(20), 13229.

Kader Esen, V., Palangi, V., & Esen, S. (2022). Genetic improvement and nutrigenomic management of ruminants to achieve enteric methane mitigation: A review. Methane, 1(4), 342-354.

AU: All suggested citations were added to the revised manuscript (Lines 720-725), but it is worth noting that the references of the authors’ works are highly relevant to the topic of this manuscript and were used in several parts of the manuscript. Therefore, we would suggest to keep them in the manuscript.

Reviewer#2:

1. Grammar corrections.

AU: Thank you for your suggestions. All corrections were made within the revised text as suggested (Lines 29, 253, 603).

2. Please include a reference for this sentence

AU: Reference was added as recommended (Line 43).

3. Please split this into a new paragraph here.

AU: Reference was added as recommended (Line 49).

4. REML and Bayesian methods are estimation frameworks, whereas the animal model is a model structure that can be fitted using either approach. Consider rephrasing for clarity.

AU: Change made as suggested (Lines 126-128).

5. This seems redundant, as it has already been described in the M&M section. Please consider removing it.

AU: It was removed as suggested.

6. Please use a maximum of three decimal places

AU: Change made as suggested (Line 242).

7. I suggest ordering the table by chromosome position.

AU: Change made as suggested (Table 3).

8. could you explain what Weight means?

AU: “Weight indicates the total sample size contributing to the meta-analysis for each SNP, corresponding to the sample-size–weighted Z-score approach implemented in the METAL program”. This information was added to the footnote of Table 3.

9. Could you explain what the different colors in the figure mean?

AU: Node color intensity reflects the significance of GO term enrichment (adjusted P-value), with darker colors indicating more significant enrichment (Lines 320-322).

10. Consider not repeating in the text the values that are already clearly shown in the table, to improve readability. Please check it out in this section.

AU: Changes made as suggested (Lines 391-402, 422, 430, 436).

11. I suggest to include below the table the descriptions of some of the column names

AU: Changes made as suggested (footnotes of Tables 6 and 7).

12. Since heritability is constrained to be non-negative, it is unclear why the confidence intervals in the figure extend below zero. Please provide clarification or justify the estimation approach that leads to this.

AU: Some explanations were added to the text for clarification as recommended (Lines 620-625).

13. Please rewrite this sentence and avoid including results (estimated values). Try to focus on discussion and comparison of the results.

AU: Changes made as suggested (Lines 610-613).

---

## [Editor Report · Decision Letter 1]

24 Feb 2026

Genomic dissection of methane emission traits in cattle: a meta-GWAS and heritability analysis across populations

PONE-D-25-57330R1

Dear Dr. Ghavi Hossein-Zadeh,

We’re pleased to inform you that your manuscript has been judged scientifically suitable for publication and will be formally accepted for publication once it meets all outstanding technical requirements.

Kind regards,

Hayrunnisa Nadaroglu

Academic Editor

PLOS One
---

## [Editor Report · Acceptance letter]

PONE-D-25-57330R1

PLOS One

Dear Dr. Ghavi Hossein-Zadeh,

I'm pleased to inform you that your manuscript has been deemed suitable for publication in PLOS One. Congratulations! Your manuscript is now being handed over to our production team.

Kind regards,

on behalf of

Professor Hayrunnisa Nadaroglu

Academic Editor

PLOS One